# A Design of Electromagnetic Velocity Sensor with High Sensitivity Based on Dual-Magnet Structure

**DOI:** 10.3390/s22186925

**Published:** 2022-09-13

**Authors:** Xiao Zhou, Yangfan Ruan, Xingang Mou, Yuhao Yuan, Yi He

**Affiliations:** 1School of Mechanical and Electronic Engineering, Wuhan University of Technology, Wuhan 430070, China; 2Intelligent Transport Systems Research Center, Wuhan University of Technology, Wuhan 430063, China

**Keywords:** dual magnet, velocity sensor, sensitivity, magnetic field intensity

## Abstract

The most permanent magnets in current electromagnetic velocity sensors are magnet cylinders that have been axially magnetized, with magnetic boots changing the propagation direction of the magnetic induction lines of the magnet cylinders. However, the magnetic field generated by the magnet cylinders is not fully utilized, which leads to uneven magnetic field intensity of the working air-gap and high magnetic field intensity of the nonworking air-gap. We propose a novel dual-magnet structure (DM) mainly consisting of two magnet loops that are magnetized radially and a magnetic conductive shaft, adopting a concentric nested configuration. The dual-magnet structure can make the magnetic induction lines enter the working air-gap directly from the magnet and increase the effective magnetic field, which is perpendicular to the coils in the working air-gap. This design can further improve the sensitivity of a velocity sensor and enhance its ability to detect weak signals in microtremor exploration. The validity of the dual-magnet structure has been established by numerical simulations and verified by experiments. The results reveal that the magnetic field intensity is increased by 29.18% and the sensitivity is improved by 23.9%, when the total volume and material of the magnet are unchanged. The full utilization of the material is achieved without increasing the complexity of the structure.

## 1. Introduction

Microtremor exploration is a novel and essential engineering method to obtain ground motion signals. It finds use in urban geothermal exploration [1], buried fault structures detection [2], structural health monitoring (SHM) [3,4], seismic site-specific characteristics estimation [5], and other engineering applications [6]. The conventional exploration method utilizes the underground medium’s reflection or refraction at the underground’s wave impedance interface to obtain geological information. In cities or other areas with complex geological conditions, boreholes cannot be implemented due to underground pipelines and buildings, leading to the loss of specific geological information. Given the disadvantage of high cost, the traditional geophysical exploration methods are severely limited.

On the other hand, microdynamic exploration, which is non-invasive and environmentally friendly, uses environmental or artificial noise and can effectively measure shear wave velocities at many stations in dense urban areas [7,8]. It has excellent potential in field tests [9] and is expected to become a new noninvasive exploration method with a wide range of potential applications. Figure 1 shows a common application of microtremor exploration. According to the ground’s features and measuring methods, different arrangements of arrays are selected. The wavenumber–frequency method (f–k) and spatial autocorrelation method (SPAC) are common and effective microtremor survey methods [10]. A vast number of array arrangements are reported in case histories such as cruciform, hexagon, and triangle [11]. The f–k method usually needs more sensors and has more flexible arrangements of arrays, such as the geometry of cruciform, while the SPAC method usually requires the sensor to be placed in hexagon or triangle arrays. In the industrial field, the array arrangements of sparse nested or common-base triangles with the SPAC method are widely used for their advantage of offering sufficient azimuthal averaging in most cases [12,13]. The natural field source microtremor signals are detected by sensors in the array. Then, geological information is obtained by extracting dispersion curves and analyzing the relationship between phase velocity and depth.

The selection of sensors in microtremor exploration is based on sensitivity, stability, durability, and economics [14]. With the advantages of being light, durable, economical, and self-powered, the electromagnetic velocity sensor is one of the most effective sensors in ground motion measuring [15]. Since microtremor exploration uses natural noise with weak signals, it is particularly sensitive to solid vibration noises in surroundings. In addition, under normal conditions, the magnitudes of microvibration signals are very small [16] (i.e., −3 to 1 MW), and their effective frequencies usually range from a few Hz to several hundred Hz [17,18], which places higher demands on the instrument’s ability to detect weak signals. Sensitivity is an important parameter in evaluating the ability of the sensor to respond to vibration signals. In recent years, the production of low-frequency sensors with reliable sensitivity has been one of the main goals for researchers [19]. 

Mawa Patrick Luka et al. [20] analyzed the phase hysteresis, resistivity, and surface current density of electromagnetic velocity sensors. They studied the linear damping and sensitivity of electromagnetic velocity sensors at low frequencies, designing and producing low-frequency velocity sensors with different damping ratios. The conclusion is that the sensitivity reaches its highest when the damping ratio is 75%. Dennis Ling [21] studied the sensitivity of velocity sensors degrading over time. By totaling an external resistor, an experimental study was conducted to improve the current distribution caused by phase lag. They found that the amplitude of output voltage can be increased when a resistor of 500 Ω is added. Anastasia Fokina [22] proposed a mathematical model and an algorithm for temperature compensation, which shortened the operation time of device setup. Jiheng Ding [23] developed an active vibration device with an adjustable electromagnetic negative stiffness, which provides a low linear composite stiffness in a stroke range of ±4.4 mm, and the sensitivity is 275 V/m/s.

Fan Xiaoyong et al. [24] designed fixtures for horizontal and vertical shake tables. By using a dial to obtain a more accurate azimuth, the high-precision measurement of cross-sensitivity for low-frequency sensors is satisfied. Li Hong et al. [25] designed a closed magnetic field structure, which increases the magnetic field intensity by 24%, in comparison with the conventional semiclosed structure. At the same time, the uniform range of the magnetic field and the output voltage of the coil are improved.

However, previous studies have mainly focused on the discussion of influencing factors of sensitivity. In addition, to significantly increase the sensitivity of velocity sensors, other structures or techniques are needed. In ref. [23], the sensitivity of the vibration system reaches as high as 275 V/m/s. However, apart from two extra magnet loops and coil winds, the active control of drift suppression and a force balance are also combined. This would lead to bigger volume, greater mass, and higher design and operational complexity. Moreover, the magnetic fields of magnet cylinders are mostly magnetized axially. The use of magnetic conductive materials to change the direction of magnetic flux would lead to problems such as hysteresis, uneven distribution of magnetic induction lines, and severe edge effects [26]. Further analysis and optimization of the magnetic field structure design will help to improve the overall performance of the sensor, especially sensitivity [27]. 

In this paper, we propose a novel dual-magnet structure with radial magnetization, which reduces the hysteresis, increases the uniformity and density of the magnetic field, and effectively improves the sensitivity. A detailed investigation of edge effects is beyond the scope of this work. We acknowledge that further studies are required to clarify this question. So, it will not be the emphasis of this paper.

In Section 2, the principle of electromagnetic velocity sensors and the relationships between sensitivity and magnetic field intensity are discussed in detail. In Section 3, a 3D model of the dual-magnet velocity sensor is introduced, and a simulating analysis of it is completed. Section 4 is the experimental result of this paper, where the effect of the dual-magnet structure on the magnetic field and the improvement of sensitivity are verified. In Section 5, the future work of this study is discussed. Section 6 is the summary and conclusion of the whole research.

## 2. Analytical Model and Influencing Factors

### 2.1. Principle of Electromagnetic Velocity Sensor

Velocity sensors are widely used as vibration detectors. Based on modern control theory [28], velocity, displacement and other state variables can be utilized in the feedback. Compared with accelerometers, velocimeters derive the displacements mainly by one integration, which would eliminate the two integration errors caused by the drift of the acceleration sensors in the low-frequency region [29].

In most cases, the electromagnetic velocity sensor is based on a moving-coil structure, with electromagnetic induction as the working principle. The sensor can be simplified as a spring-mass-damper oscillator, as indicated in Figure 2. The acceleration of the shell of the sensor exerts an inertial force on the magnet. On the one hand, the permanent magnet is fixed in the shell and vibrates with it when the movement of the ground occurs. On the other hand, the coil is connected to the shell by an elastic spring that can move up and down in the vertical direction inside the sensor. Due to inertia, when the vibration frequency of the object improves, the coil cannot keep up with the vibration of the external objects and remains stationary in absolute space. Therefore, there will be relative motion between the coil and the magnet. By cutting the magnetic induction lines, the coil outputs induced voltage. The relative motion between the coil and the shell is an objective reflection of the actual ground motion function [19].

The induced voltage is defined by Equation (1) [30],
(1)$$U=B_{i}L_{0}N_{i}v_{0},$$
where $B_{i}$ is the magnetic field intensity of the working air-gap (in T), $L_{0}$ is the average length of each turn of the coil (in m), $N_{i}$ is the number of turns of the coils in the working air-gap, and $v_{0}$ is the velocity of the vibrating object (in m/s). Figure 3 illustrates the physical implications of each variable. As indicated in Figure 3, there are two types of air-gaps: the working air-gap and the nonworking air-gap. The working air-gap refers to the motion scope of the coils, where the magnetic induction lines are cut and induced voltage is produced, while the nonworking air-gap is out of the reach of any coils. Therefore, to further improve the value of the induced voltage, it is wise to increase the magnetic field intensity in the working air-gap and decrease the magnetic field intensity in the nonworking air-gap.

### 2.2. Effect of Sensitivity on Output Signal

For sensors, sensitivity is the ratio between the output voltage and the vibration velocity, which represents its ability to convert the movement of the ground to corresponding electrical signals. Therefore, sensitivity is one of the most important performance parameters of electromagnetic velocity sensors [31].

The sensitivity of an open-loop electromagnetic velocity sensor is given as follows [32],
(2)G=Bl,
where G stands for sensitivity, B is assumed to be the effective magnetic field intensity, and l is the total effective length of the coils. Given a damping ratio of 0.707 as well as a natural frequency of 4.5 Hz, the frequency characteristic curves for different sensitivities are depicted in Figure 4.

As can be seen from Figure 4, by taking different sensitivities of G = 5, G = 10, G = 20, G = 50, and G = 100, the phases of output signals remain unchanged. In addition, the amplitudes of the output signals are positively correlated with sensitivity. As a key parameter of indicating the ability to receive weak vibration signals, the sensitivity should be higher as the intensity of the vibration signals is lower. Therefore, the sensitivity is expected to be increased as much as possible when a sensor is designed, so that the received signals can be more easily separated from the noises.

### 2.3. Effect of Magnetic Field Direction on Sensitivity

Equation (1) provides the truth that the sensitivity is mainly related to the magnetic field intensity and the total length of the coils. The larger the magnetic field intensity and the longer the total length of the coils, the higher the sensitivity will be. However, as coils are lengthened, the mass and volume of the sensor will increase correspondingly. In comparison, it is more feasible to improve the magnetic field intensity.

There are two types of magnetization directions for a magnet loop: axial magnetization and radial magnetization. As shown in Figure 5, blue indicates the S-pole, and red indicates the N-pole. Figure 5a,b present the two different magnetization directions of the magnet loops, and Figure 5c,d plot their corresponding cross-sectional views and label the molecular currents and magnetization directions. The yellow arrows are the current directions on different surfaces of the magnet loops, and $\vec{M}$ indicates the magnetization direction. For an axially magnetized magnet, the magnetic field is produced by the molecular currents on the inner and outer sides of the loop [16], where the current directions are opposite. For a radially magnetized magnet, the magnetic field is produced by the molecular currents on the upper and lower surfaces of the loop, where the current directions are also opposite. To gain more insight, the best magnetization direction of a magnet loop is demonstrated below.

Take N as the number of turns of coils moving relatively to the magnet in the velocity sensor. Its output voltage is given by Equation (3) [33],
(3)$$\varepsilon =-N\frac{d\phi }{dt},$$
where ε stands for the value of output voltage, N is the number of the coils in the working air-gap, and $\frac{d\phi }{dt}$ refers to the induced voltage produced by a single turn of the coil.

Take one of the turns of the coils for analysis. There are magnetic induction lines of different directions cutting the coil to generate the induced current. A small section of the coil is randomly selected. 

As is illustrated in Figure 6, $t_{0}$ is assumed to be the initial moment, and $x_{0}$ is assumed to be the initial position of the current; after a small period, it moves to position $x_{1}$, at the moment of $t_{1}$. $S_{1}$ and $S_{2}$ are assumed to be the two end surfaces of the coil. In addition, $S_{3}$ is assumed to be the side surface of the coil section. The direction vector of the $S_{1}$ surface is $k_{1}$. The direction vector of the cylindrical surface formed by the area $S_{1}$ and the side surface $S_{0}$ is $k_{2}$. Equation (4) describes the feature of magnetic flux in the coil [34],
(4)$$\int B\cdot dS=\int B\cdot k_{1}dS_{1}+\int B\cdot k_{1}dS_{0}+\int B\cdot k_{2}dS_{3}=0,$$

The side surface can be calculated as Equation (5),
(5)$$S_{3}=2\pi r\Delta x,$$
where r is set to be the radius of an end surface, Δx is the length of the coil section. When time ($t_{1}-t_{0}$) approaches zero, Δx equals dx.

The output voltage can be further expressed in the form of Equation (6),
(6)$$\varepsilon =-(\int B\cdot k_{1}dS_{0}+\int B\cdot k_{1}dS_{1}),$$

Equations (4) and (5) are substituted into Equation (6), then the voltage generated in the coil can be obtained as Equation (7),
(7)$$\varepsilon =2N\frac{d}{dt}\int B\cdot k_{2}\pi rdx=2N\frac{d}{dt}\int_{x_{0}}^{x_{1}}Bsin\theta dx,$$

When the induced electric potential generated in the coil is maximized, the sensitivity reaches its top. In this case, sinθ=1. The direction of the magnetic field in the air-gap is perpendicular to the direction of the motion of coils. In addition, there are no x-direction components. For electromagnetic velocity sensors, there is a minimum x-direction component (zero) of the magnetic field, when an axially magnetized loop is selected. Therefore, compared to axial magnetization, radially magnetized magnets are more effective in increasing the magnetic field intensity in the air-gap and improving the sensitivity and output voltage of the sensor.

## 3. Structure Design and Numerical Simulation

### 3.1. Mathematical Model of Dual-Magnet Velocity Sensor

There is also some previous research about magnet loops [35,36], which introduced an innovation in modifying the physical structure of velocity sensors. Our design is based on these studies. The distinction is that we changed the materials of magnets and optimized the size and structure with FEA (finite element analysis). Figure 7 illustrates the structure of a conventional low-frequency velocity sensor, which is mainly made up of a coil, a coil frame, two spring diaphragms, a magnet cylinder, and two boots. The magnet cylinders of a conventional low-frequency velocity sensor are axially magnetized, and the magnetic boots made of pure industrial iron are attracted on the upper and lower sides of the magnet cylinder, which is used to change the direction of the magnetic induction lines, making it disperse from the vertical direction to the horizontal direction. 

Figure 8 illustrates the 3D model of the dual-magnet velocity sensor. The magnet loops and the coils, which are wound on the coil frame, form an electromagnetic induction system. The magnet loops, spring diaphragm, and shell form a mechanical system. In addition, the magnetic field generated by the magnet loops is enclosed in the shell. The coil is connected to the shell by a spring diaphragm and is fixed in the magnetic field, serving as an inertial mass moving relative to the magnet loops. The bottom of the sensor is coupled to the ground. The magnet is magnetized in the radial direction, with the upper and lower pieces magnetized in opposite directions and connected by a magnetic conductive shaft in the middle. 

The magnetic conductive shaft is made of pure industrial iron with low magnetic reluctivity. Accordingly, the major advantage of the shaft is the excellent magnetic property. Figure 9 presents the structure of the magnetic conductive shaft and its assembling relationship with other components. It can be seen that the shaft is not simply in the shape of a cylinder but is designed to be thick in the middle and thin at two ends, which can ensure the fix of the magnet loops under the attraction force between them. The magnet loops are coaxially mounted with the shaft and the pure copper spacers close to them, where electronic signals can go through, and the magnets are fixed. The difference is that for the DM velocity sensor, there is no need to change the direction of the magnetic field in the air-gap with other materials. This can reduce the damping of magnetic field intensity and improve the uniformity of it. Note that there exists a gap between the ends of magnetic conductive shaft and the copper spacers, to ensure that their end faces can form a tight bond.

### 3.2. Magnetic Field Structure of Dual-Magnet Velocity Sensor

Finite element analysis software (ANSYS Electronics Suite 2022 R1) is used to simulate the magnetic field intensity in the dual-magnet velocity sensor. First, a Maxwell 2D model with components that had been assigned corresponding materials was created. Second, the boundary (vector potential equaled zero) and excitation (permanent magnet field) were defined. Then, in the solution setup, the maximum number of passes was set to be 10. The range of magnetic field intensity is from 0 to 1200 mT. Finally, the contour map and vector plot were obtained in Figure 10. It can be seen from Figure 10a that the magnetic field intensity in the air-gap is between 450 mT and 700 mT, while the magnetic field intensity in the nonworking area is between 0 and 72 mT. Figure 10b indicates that both magnet loops are radially magnetized, but the direction of the magnetization is opposite.

Additionally, the loops, magnetic conductive shaft, shell, air-gap, and magnet loops form a complete magnetic circuit. It should be pointed out that the total volume of the two magnet loops added together is equal to the volume of the magnet in a conventional model, as can be seen in Figure 11. In addition, the same materials of permanent magnets were used, both of which were NdFeB magnets (Table 1). The advantage of this structure will be discussed in detail in Section 4.

As is detailed in Figure 12, Figure 12a shows the magnetic flux distribution of the dual-magnet structure, while Figure 12b reveals the magnetic flux distribution of the conventional structure. It can be found that the magnetic induction lines of the DM sensor are dense and even, while the magnetic induction lines of the traditional one are sparse. In addition, the magnetic induction lines going through the top of the magnetic boots are deflected greatly. Therefore, the magnetic field homogeneity of the DM sensor is better than that of the conventional structure. Note that in Figure 12a, the magnetic conductive shaft is coaxially coordinated with a copper loop, which features low permeability ($1.26\times 10^{-6\;}H/m$) [38] and high electrical conductivity ($5.96\times 10^{7\;}S/m$) [39]. On the one hand, it prevents the magnetic induction lines from returning directly, without reaching the next magnet. On the other hand, it has a conductive effect on the induced currents.

### 3.3. Circuit Structure of Dual-Magnet Velocity Sensor

Figure 13 presents the internal circuit of the dual-magnet velocity sensor. The circuit loop mainly consists of two copper spacers, two magnet loops, two spring diaphragms, a magnetic conductive shaft, a coil frame, and the coils. In Figure 13b, two common circuits are plotted, and it is important to highlight that the two magnet loops do not set off against each other. Instead, they obey the linear superposition principle. All the currents follow their respective directions, ultimately outputting the electrical signals through the pins on the upper cover. Inside the coil frame, as labeled with red arrows in Figure 13b, the induced currents go through the following components in turns: the upper copper spacer, the upper magnet loop, the magnetic conductive shaft, the lower magnet loop, the lower copper spacer, and the spring diaphragm. Outside the coil frame, there are two sets of enameled wires connected in series. Both of them are welded to spring diaphragms, as can be seen in Figure 13a. Between the upper spring diaphragm and the copper cap, a plastic membrane is added as an insulating layer to prevent short circuits.

As Figure 14 shows, the winding of the coils has the following characteristics: the upper and lower sets are wound in opposite directions and in series with each other. Moreover, the directions of magnetic induction lines of these two places are opposite. The advantages of this design are as follows.
It can be explained by Ampere’s right-hand screw rule that, when the direction of the magnetic field and winding coils in the upper and lower working air-gaps are both opposite, the inductive electric potentials produced, respectively, by the two sets will be in the same direction. Therefore, the total inductive voltage will become promoted.According to Maxwell’s equations, when the coil moves relative to the magnetic field at a changing speed, the induced current will change consequently, which, in return, excites an extra magnetic field in space. This will influence the stability of the magnetic field generated by the magnet loops. By designing two sets of coils with the same number of turns and opposite winding directions, they can cancel each other out by exciting magnetic fields of the same magnitude but opposite directions in space, improving the system’s anti-interference capability.

## 4. Experimental Comparison and Analysis

To confirm the simulation results, experiments are conducted to measure the magnetic field intensity and sensitivity of velocity sensors. Another sensor with a magnet cylinder that is magnetized axially is selected as a comparison.

### 4.1. Measurement of Magnetic Field Intensity

#### 4.1.1. Experimental Methods

The actual picture of the DM sensor is displayed in Figure 15. For a magnet loop, the internal diameter is 5 mm, while the outside diameter is 18.80 mm. The height of it has been set to be 5.35 mm. The final dimensions of the DM sensor are ∅25.4 mm × 36 mm.

To confirm the validity of the design, the magnetic field homogeneity and magnetic induction intensity of the two magnetic field structures were tested. One is the PS4.5-C1 velocity sensor (PS), with a magnet that is magnetized axially. The other one is the dual-magnet velocity sensor (DM), with a magnet that is magnetized radially. The PS sensor is produced by Weihai Sunfull Geophysical Exploration Equipment Co., Ltd. Using a high-precision digital gauss meter (PEX-233USB) combined with a Hall sensor probe, the magnetic induction intensity of the working air-gap was measured. The sensor was close to the external surface of the coil frame, with a rated movement along the axial direction with a path length of 30 mm. Take a movement speed of 0.8 cm/s. The values of the digital gauss meter at different moments were recorded to compare the magnetic field distribution of the two structures.

#### 4.1.2. Experimental Results

In Figure 16, the experimental results of the conventional magnetic field structure (PS) and the new magnetic field structure (DM) are provided. The blue dashed line shows the simulated results of magnetic field distribution of PS sensor, and the blue solid line is the measured results of magnetic field distribution of PS sensor. The red dashed line presents the simulated results of magnetic field distribution of DM sensor, and the red solid line stands for the measured results of magnetic field distribution of DM sensor. The simulated results are calculated and analyzed with electromagnetic simulation software. The horizontal coordinate is a path (in mm), and the vertical coordinate is magnetic field intensity (in mT). The experimental results indicate that the magnitude of magnetic field intensity has double peaks. The area with high magnetic field intensity matches the range of movement for the sets of coils in the sensor, while the nonworking area is with low magnetic field intensity. Take the value of 90% of the maximum intensity as the lower limit for the uniform magnetic field. The maximum magnetic field intensity of the conventional magnetic field structure can reach 497 mT. Its working space is from 5.2 mm to 8.4 mm and from 18.9 mm to 22.1 mm. The corresponding length is 3.2 mm for each set of coils. For the new magnetic field structure, the maximum magnetic field intensity is 642 mT, with working space from 5.0 mm to 9.2 mm and from 18.0 mm to 23.2 mm. The corresponding length is 4.2 mm, where the magnetic field is relatively uniform. 

Therefore, it can be known that the magnetic induction intensity in the working air-gap is increased by 29.18%, and the magnetic field uniformity is improved by 31.25%, compared with the conventional one. There is a discrepancy between the measured results and the simulated results. This may result from the difference in material properties and processing precision [25]. Alternatively, the sensor is not fully enclosed during the movement of the Hall sensor probe, resulting in the leakage of several magnetic induction lines. However, for the PS sensor, the maximum difference of magnetic induction at the same position is within 40 mT, with a relative error of less than 7.5%. In addition, for the DM sensor, the maximum magnetic induction at the same position is within 60 mT, with a relative error of less than 8.4%. Both of them are acceptable. As illustrated in Figure 16, the magnetic field structure of axial magnetization makes the magnetic field strength and magnetic field uniformity in the working air-gap improved.

### 4.2. Measurement of Sensitivity

#### 4.2.1. Measuring Principle

The DC excitation method was chosen in the laboratory as a method to test the sensitivity of the dual-magnet sensor, which is a simple and effective way to determine the dynamic parameters of the sensor and is widely used by researchers and physical prospectors [40].

The principle diagram of the test is offered in Figure 17.

The test system consists of a DC supply, a limiting resistance, a digital multimeter, an oscilloscope, a computer, and a DM sensor. The experimental principle and its operation are as follows. When the sensor is connected to the circuit, and DC excitation is applied, the coils of the sensor will be subjected to an electrodynamic force and deviate from the equilibrium position. After the coils are stabilized, the excitation current is momentarily disconnected, and the coils will undergo a decaying motion in the original equilibrium position. Therefore, the coils would cut the magnetic induction lines and output voltage signals. An oscilloscope was used to capture the output voltage waveform, and the corresponding data were processed by a computer. As is reflected in Figure 18, the sensitivity of the velocity sensor can be derived from the characteristics of this waveform, combined with Equations (6)–(9) (*A*_1_ is the first peak, *A*_2_ is the second peak, and *T*_0_ is the first moment when the voltage is zero) [41].

#### 4.2.2. Results of Test

To provide a suitable magnitude of stable direct current, a series resistance of 10 KΩ is added to the circuit. The instruments of the test system are presented in Figure 19. Increase the excitation current by gradually adjusting the output voltage. Take $4\times 10^{-4}$ s as the sampling interval, and take 2500 sampling points every time. The 1250th point is set as the sampling point. Both the sampling point of the DM sensor and PS sensor are tested. Figure 20 represents the graph captured by the oscilloscope.

The velocity sensor voltage response has the form as follows [40],
(8)$$y\left(t\right)=-\frac{GI_{0}}{m}\cdot \frac{\exp\left(-\eta_{0}\omega_{0}t\right)}{\omega_{0}\sqrt{1-\eta_{0}^{2}}}\sin\left(\omega_{0}\sqrt{1-\eta_{0}^{2}}t\right),$$
where G represents the sensitivity of the sensor, $I_{0}$ is the excitation current, m is taken as the inertial body mass, $\eta_{0}$ is the damping factor, and $\omega_{0}$ stands for natural frequency ($\omega_{0}=2\pi f_{0}$).

Where y(t) equals to $A_{1}$, the sensitivity of velocity can be expressed as Equation (9),
(9)$$G=\sqrt{\left(\frac{2\pi f_{0}mA_{1}}{I_{0}}\right)\cdot \exp\left(\frac{\arctan e}{e}\right)},$$
where $e=\sqrt{1-\eta_{0}^{2}}/\eta_{0}$.

Equation (9) suggests that the sensitivity is positively correlated with $A_{1}$. As is reflected in Figure 20, due to the difference in overall resistance, the initial voltages of these two sensors are also different. However, it does not matter as long as the magnitude of the currents in the corresponding experimental group (with DM sensor) and the control group (with PS sensor) remain the same. It can be seen that the value of $A_{1}$ in the experimental group is much higher than that in the control group, which indicates that the DM sensor can have higher sensitivity.

The size of the excitation current can be referred to as Equation (10),
(10)$$I_{0}=\frac{m\omega_{0}^{2}x}{G}=\frac{4\pi^{2}mf_{0}^{2}x}{G},$$

x stands for the displacement of the coil in the vertical direction. When x takes the maximum displacement in the vertical direction of $x_{max}$, G is the sensitivity of the sensor. The maximum excitation current can be rewritten as Equation (11),
(11)$$I_{0}=\frac{4\pi^{2}mf_{0}^{2}x_{max}}{G},$$

For a PS sensor, the natural frequency is 4.5 Hz. The damping ratio is 0.76. In addition, the moving mass is 11.3 g. The sensitivity is 92 V/m/s. The maximum coil displacement is 4 mm. Therefore, the maximum excitation current can be derived from Equation (10) as 0.393 mA. The test was carried out three times for four different magnitudes of currents of 0.40 mA, 0.32 mA, 0.28 mA, and 0.20 mA. According to Equation (8), we finally obtained the sensitivity of the PS sensor as 86 V/m/s and that of the DM type sensor as 114 V/m/s. Since there exists a 10% error range in damping ratio and frequency, the error between the experimental result and the labeled result is 6.5%, which is allowed. It also proves that the operation in the experiment is reliable.

### 4.3. Comparison with Existing Velocity Sensors

The general electromagnetic velocity sensors with part of the essential parameters are listed as follows. It can be found that they have similar natural frequencies and different moving masses. The open circuit damping varies from 0.34% to 0.76%, but it does not matter, since it can be adjusted by changing the value of resistances in series with the sensors.

From the data in Table 2, the comparison of the sensitivity of different types of velocity sensors can be observed in a bar chart, as shown in Figure 21. Except for the DM sensor, the other sensors above are all magnetized axially. It is apparent from Figure 21 that the sensitivities of the first five sensors are between 28 and 92 V/m/s. It means that the sensitivities of the main commercially used electromagnetic velocity sensors are below 100 V/m/s. The sensitivity of the dual-magnet velocity sensor reaches 114 V/m/s, indicating that the sensor has a greater signal output for the same weak vibration.

## 5. Discussion

In this paper, we propose a low-frequency velocity sensor structure based on dual-magnet loops, which enhances the magnetic field strength and uniformity in the coil working space, without increasing the complexity of the system structure. The magnetic induction intensity in the working air-gap is enhanced from 497 mT to about 640 mT. By changing the direction of the distribution of magnetic induction lines in the magnets, without changing the total volume of the magnets, the sensitivity is improved from 92 V/m/s to 114 V/m/s.

Apart from sensitivity, there are also other important parameters for velocity sensors. For example, natural frequency determines the lower limit of weak signals that can be detected, which is beyond the discussion of this paper. In future research, we will further study the method of decreasing the natural frequency of dual-magnet velocity sensors. Moreover, the experiments in this research were conducted under the temperature of 26 °C. It will be useful and interesting to investigate the influence of temperature change (heating) on the device’s operating parameters.

## 6. Conclusions

Magnetoelectronic velocity sensors find great use in microtremor exploration. The sensitivity of velocity sensors is an essential factor in detecting weak signals. In this paper, without changing the total volume of the permanent magnet or increasing the system structure, the proposed structure can maximize the utilization of the magnetic field. Experimental results confirmed the effectiveness of this method. Compared to conventional structure, the magnetic induction intensity in the working air-gap is increased by 29.18%, and the sensitivity is improved by 23.9%. With slight adjustments to the proposed structure, it can fit different vibration systems with different standards. This structure has practical value in vibration detection.

## Figures and Tables

**Figure 1 sensors-22-06925-f001:**
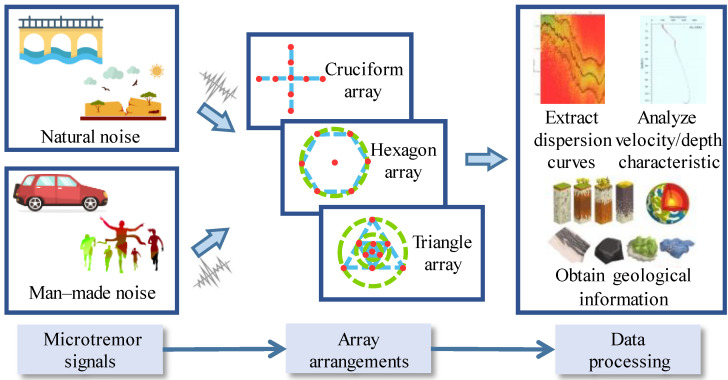
The typical applications and procedures of microtremor exploration.

**Figure 2 sensors-22-06925-f002:**
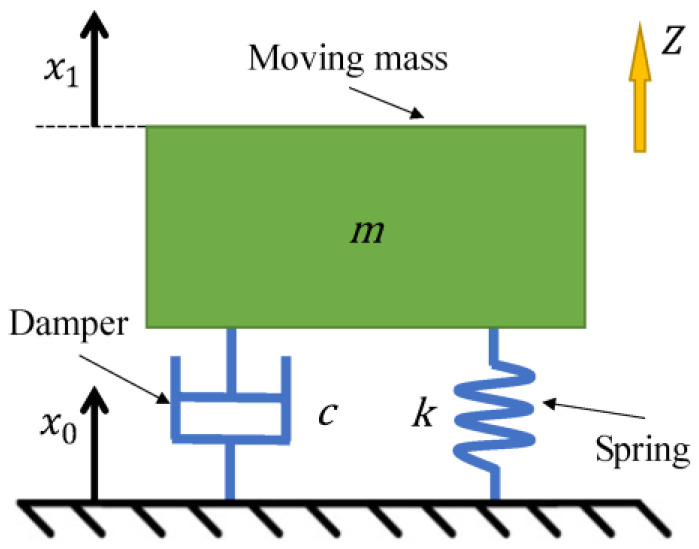
Equivalent model of electromagnetic velocity sensor.

**Figure 3 sensors-22-06925-f003:**
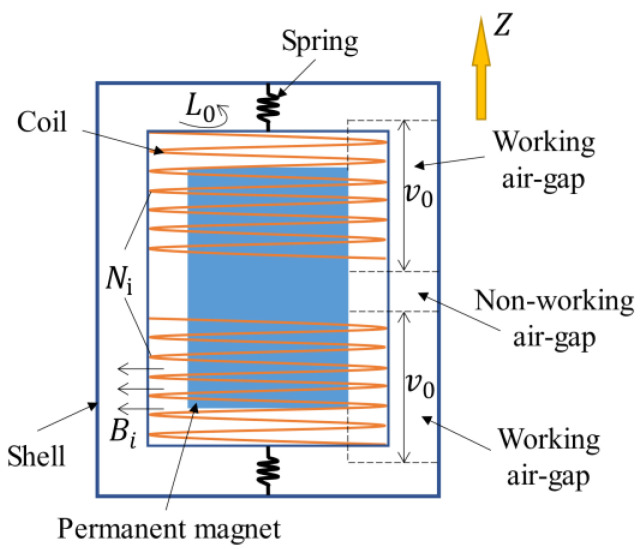
Physical implications of several essential parameters and structures in an electromagnetic velocity sensor.

**Figure 4 sensors-22-06925-f004:**
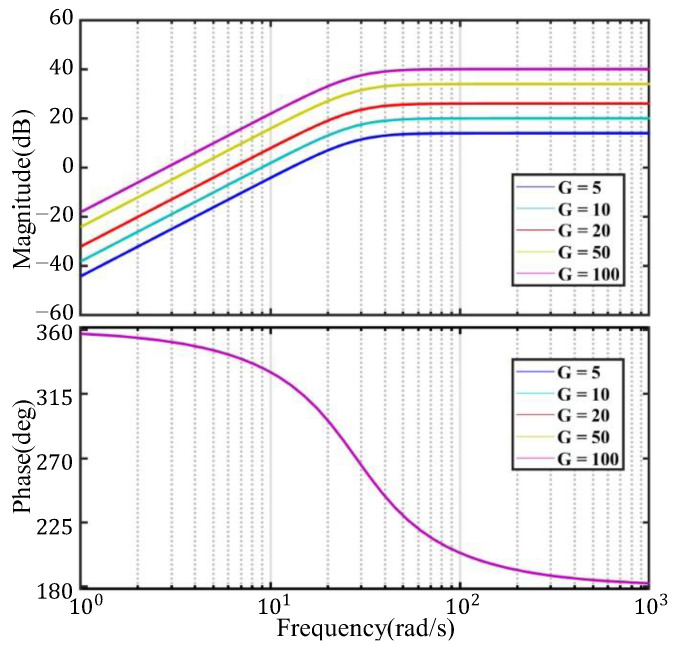
Frequency characteristic curves under different sensitivities.

**Figure 5 sensors-22-06925-f005:**
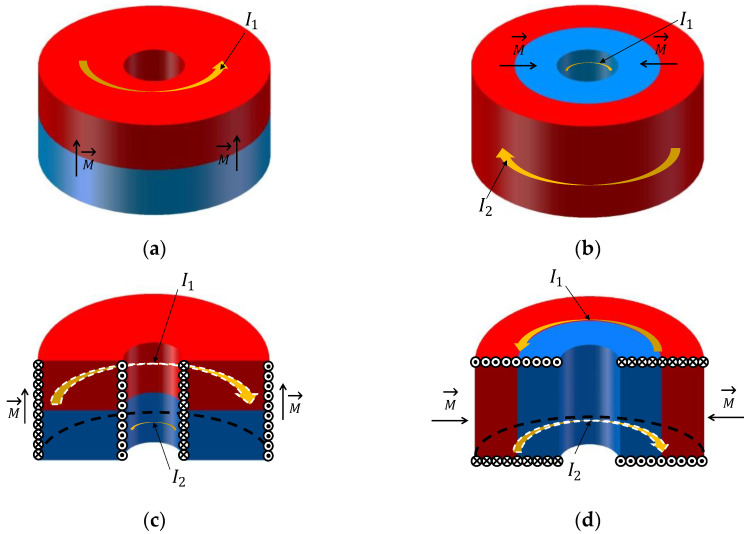
Two magnetization directions for magnet loops. (**a**) Magnet loop that is magnetized axially; (**b**) magnet loop that is magnetized radially; (**c**) molecular currents distribution for magnet loop that is magnetized radially in full-sectioned view; (**d**) molecular currents distribution for magnet loop that is magnetized radially in full-sectioned view. The solid points and fork represent opposite current-flow directions.

**Figure 6 sensors-22-06925-f006:**
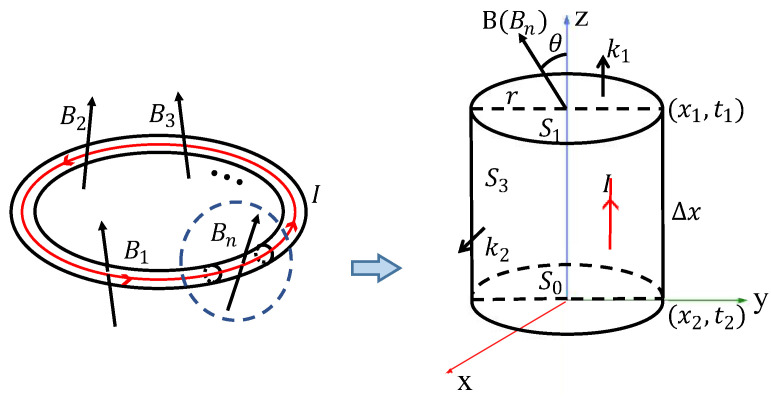
Simplified diagram of induced current in single turn coil. $B_{1}$ to $B_{n}$ represents magnetic induction lines in different directions across the coil. θ stands for one of the angles between current flow and magnetic induction lines.

**Figure 7 sensors-22-06925-f007:**
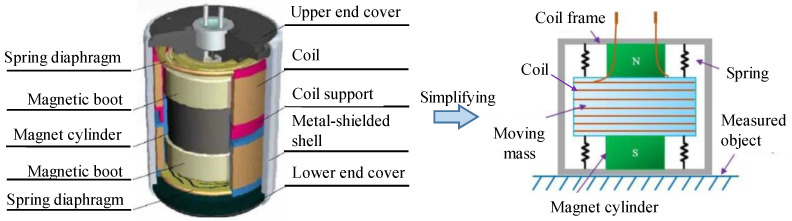
Conventional magnetoelectronic velocity sensor and its simplified model [37].

**Figure 8 sensors-22-06925-f008:**
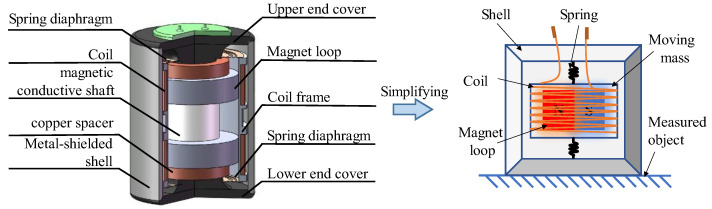
Dual-magnet velocity sensor and its simplified model.

**Figure 9 sensors-22-06925-f009:**
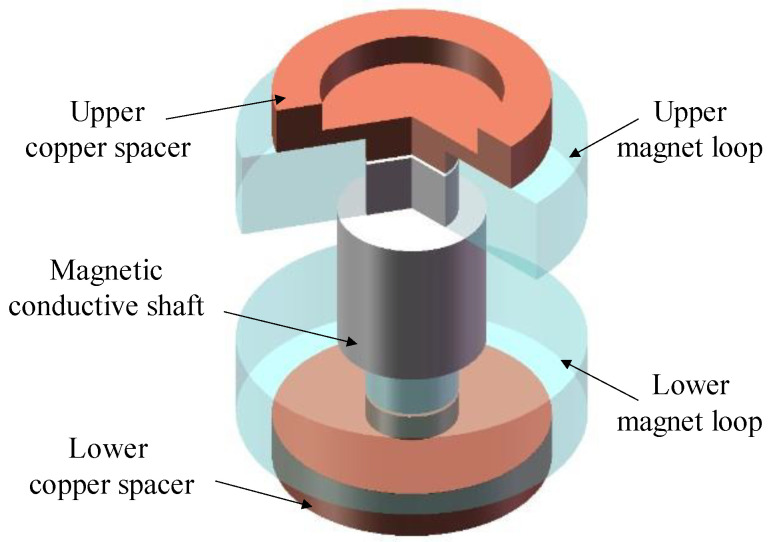
Concentric nested configuration of copper spacers, magnet loops, and magnetic conductive shaft. The model was partially sectioned.

**Figure 10 sensors-22-06925-f010:**
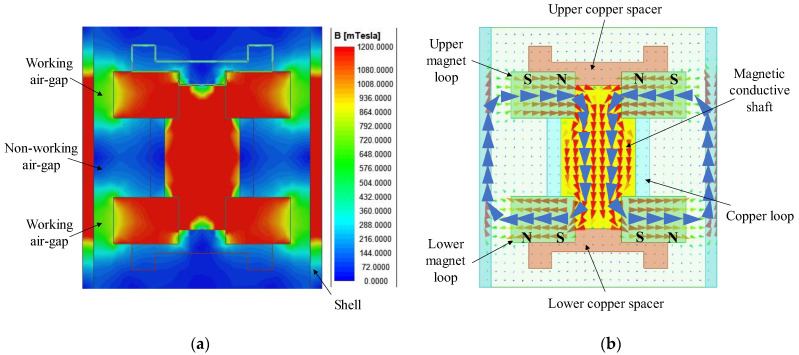
Magnetic field distribution inside the DM sensor. (**a**) Cloud diagram that reflects the intensity of magnetic field; (**b**) vector diagram that indicates the direction of magnet loop.

**Figure 11 sensors-22-06925-f011:**
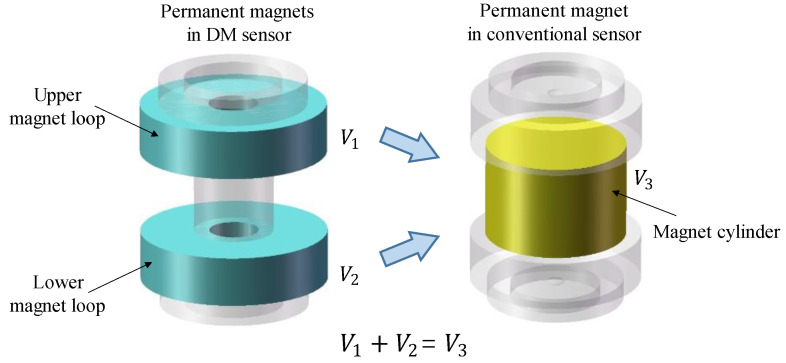
Relationship in volume size of permanent magnets between DM sensor and conventional sensor.

**Figure 12 sensors-22-06925-f012:**
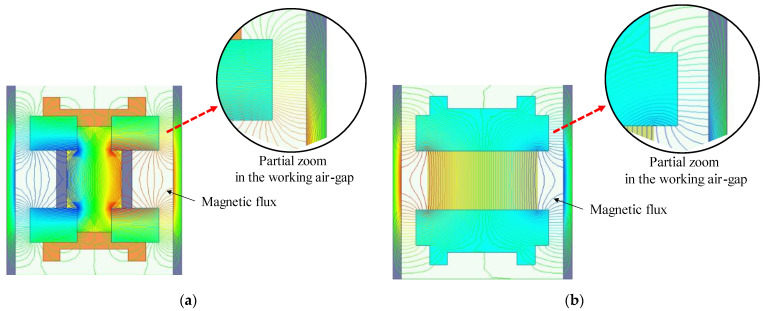
Comparison of the magnetic flux distribution of the working air-gap for the two magnetic field structures. (**a**) DM structure; (**b**) conventional structure. The density of magnetic flux depicts the intensity of the magnetic field.

**Figure 13 sensors-22-06925-f013:**
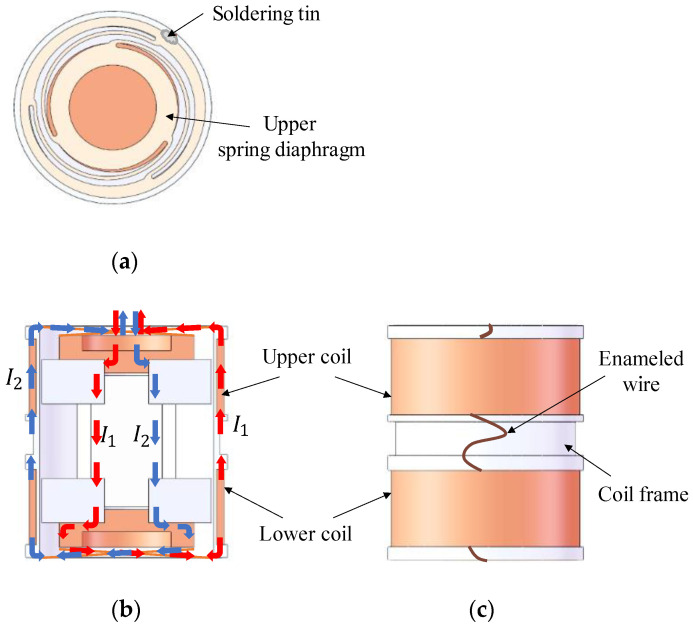
The internal structure of a dual-magnet velocity sensor. (**a**) Top view; (**b**) front view, which is full sectioned; (**c**) left view.

**Figure 14 sensors-22-06925-f014:**
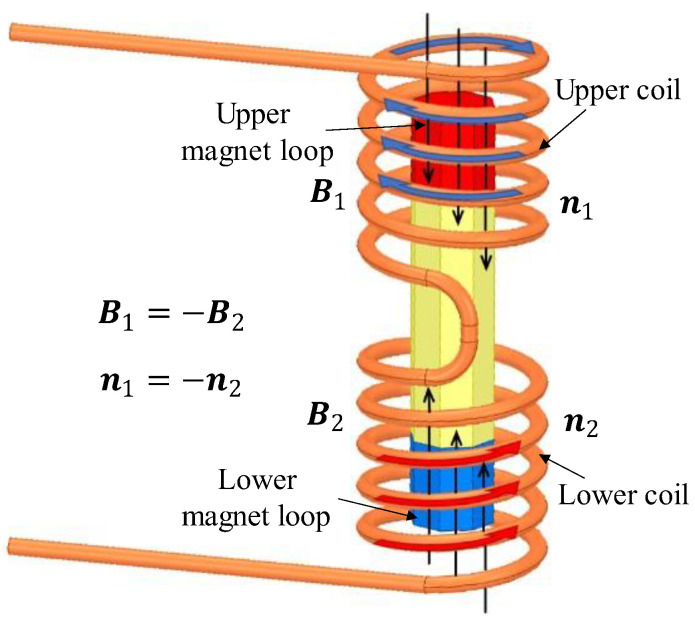
The opposite direction of the induced magnetic field. The solid red and solid blue represent two magnet loops with opposite magnetized directions.

**Figure 15 sensors-22-06925-f015:**
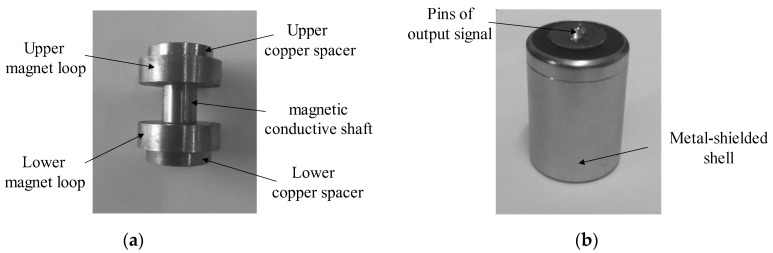
The actual picture of the DM sensor. (**a**) Assembly consisting of copper spacers, magnet loops, and magnetic conductive shaft; (**b**) outlook of DM sensor.

**Figure 16 sensors-22-06925-f016:**
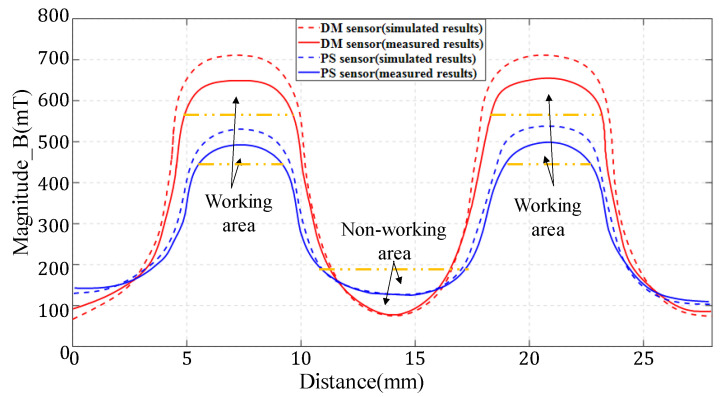
Distribution of magnetic field in air-gap of DM sensor and PS sensor. The length of the air-gap is 27 mm, and the magnetic field intensity is measured top-down.

**Figure 17 sensors-22-06925-f017:**
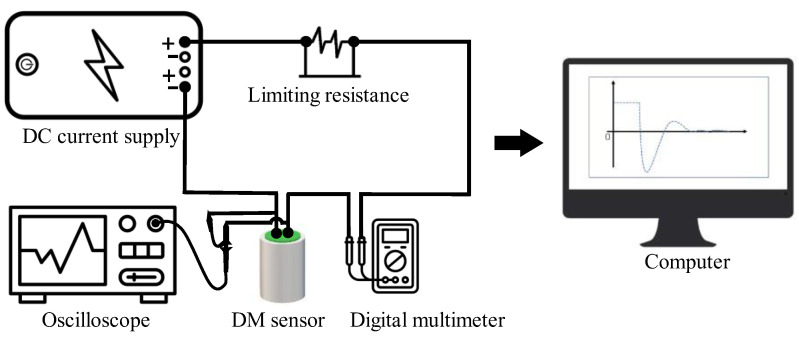
Principle diagram of DC excitation method.

**Figure 18 sensors-22-06925-f018:**
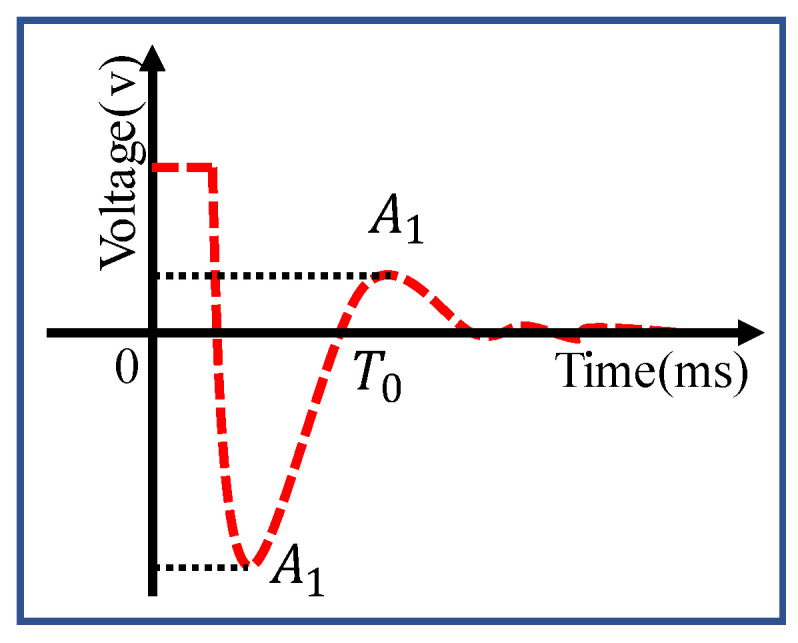
A typical attenuation curve in the DC excitation method.

**Figure 19 sensors-22-06925-f019:**
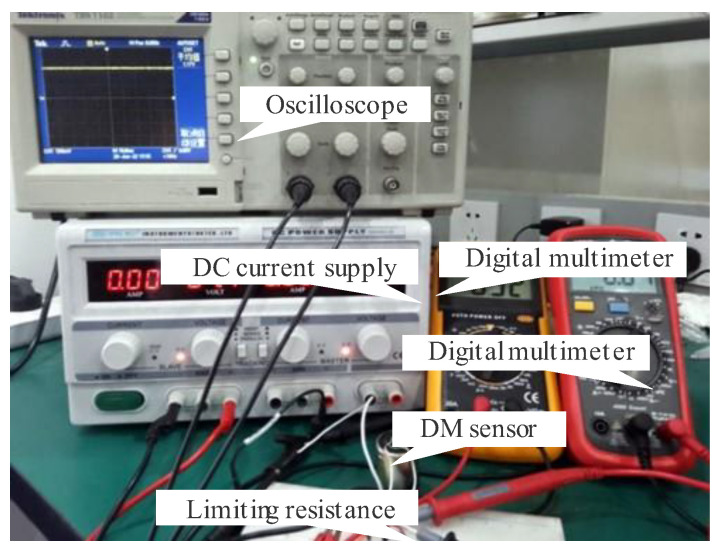
The instrument used for the DC excitation method.

**Figure 20 sensors-22-06925-f020:**
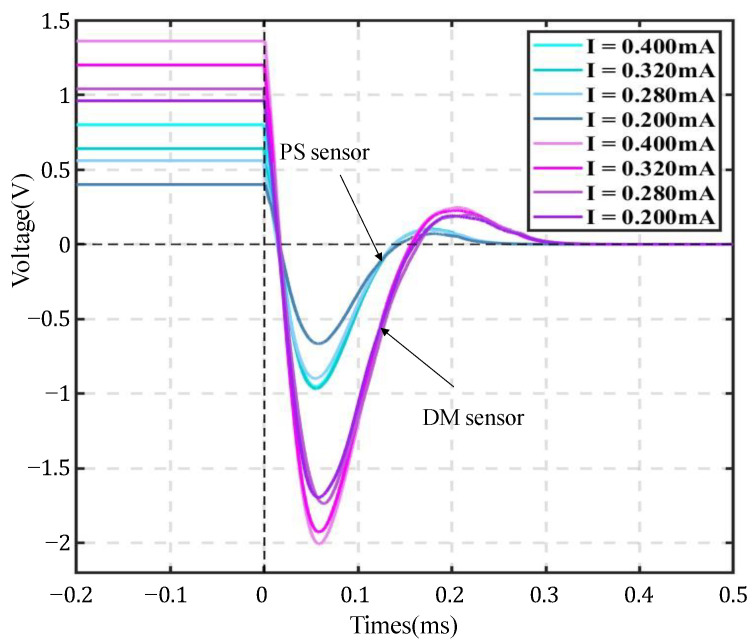
Graph of free damping motion under different excitation of currents.

**Figure 21 sensors-22-06925-f021:**
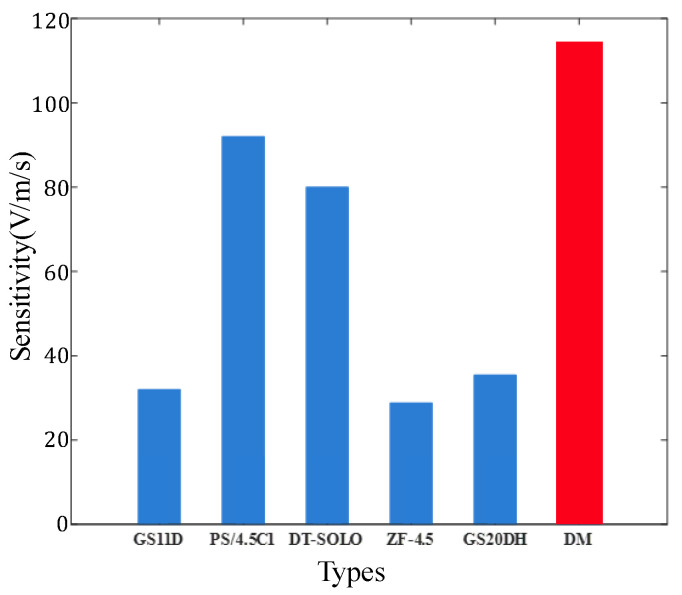
Sensitivity of different velocity sensors.

**Table 1 sensors-22-06925-t001:** Materials used in dual-magnet velocity sensor.

Parts	Materials
Magnet loops	NdFe52
Magnetic conductive shaft	Industrial pure iron
Copper spacers	Pure copper
Spring diaphragm	QBe2.0
Coil frame	Aluminum alloy
Copper loop	Pure copper
End covers	Rubber
Metal-shielded shell	Stainless steel

Parts with the same names in conventional velocity sensors and dual-magnet velocity sensors have the same materials. Magnet loops in dual-magnet velocity sensors and magnet cylinders in conventional velocity sensors both use NdFe52 materials.

**Table 2 sensors-22-06925-t002:** Comparison of the sensitivity of different types of velocity sensors.

Parameters	GS11D [42]	PS/4.5C1 [43]	DT-SOLO5 [36]	ZF-4.5 [44]	DM4.5
Natural frequency	4.5 ± 1.7% Hz	4.5 ± 10% Hz	5 ± 7.5% Hz	4.5 ± 10% Hz	4.5 ± 10% Hz
Sensitivity	32 V/m/s	92 V/m/s	80 V/m/s	28.8 V/m/s	114 V/m/s
Moving mass	23.6 g	11.3 g	22.7 g	11.1 g	11.3 g
Open circuit damping	0.34 ± 10%	0.76 ± 10%	0.6 ± 7.5%	0.56 ± 8%	0.76 ± 10%

## Data Availability

Data collected through the research presented in the paper are available on request from the corresponding author.

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
