# Peer review of "A Design of Electromagnetic Velocity Sensor with High Sensitivity Based on Dual-Magnet Structure"

_sensors, 2022, doi:10.3390/s22186925_

Round 1

Reviewer 1 Report

A new design of the velocity sensor based on dual-magnet structure is discussed in the paper. The topic is perspective and important for the industry. The great advantage of the work is that it is based on the previous researches, which was demonstrated in the introduction.

The manuscript is well structured and written in good English. The authors made good illustrations to the research. The results obtained in this paper are of interest for further research. 

However, there are some comments to the text: 

1. Authors name this device as “Magnetoelectric velocity sensor”. However, as it follows from the text, it is a new design of traditional electromagnetic sensors but not magnetoelectric. Because there is no magnetoelectric effect at all. There many magnetoelectric sensors proposed, see papers written by Prof. Dr. Nian X. Sun, Prof. Dr. Y.K. Fetisov, Prof. Dr. Eckhard Quandt etc. That is why the title of the paper is a little bit confusing. I recommend changing it to electromagnetic.  

2. Fig. 3, There is some problem with the colors. Line colors do not match legend.

3. It will be useful to indicate the directions of magnetization in Figures 4a and 4b.

4. The authors use two terms: permanent magnets and magnetic loops. It is unclear in the paper what is the difference. It should be pointed out. If there is any difference. Including the phrases like “inner and outer sides of the loop”

5. The authors use term air-gap before they show it in the figure 8. Which is not right. See line 186 for example. 

6. There is also no explanation what the “magnetic conductive shaft” is. It should be described.

7. Figure 6 does not show the coil, unlike Figure 7.

8. A comparison of the two designs only makes sense if the same materials were used. Particularly permanent magnets. There is no mention of this in the text. Actually, there is no description of any materials in the text.

Despite these comments, the article is definitely worthy of publication after small revisions. The work is interesting and useful. 

Author Response

Please see the attachments. Thank you for your time.

Reviewer 2 Report

This manuscript reports about testing of design of magnetoelectric velocity sensor. As a reviewer, I have some of the following comments and suggestions .

 1) Please check carefully the caption of Fig. 4.

 2) Please describe in more details software and measurements parameters, e.g. software version, range, step, resolution, dimensions of DM sensor, etc.

 3) The manufacturer and type of used devices should be provided.

 4) The abbreviations used in the equations (e.g. 3) should be explained.

 5) The xyz directions should be marked in the specific figures.

 6) I did not find any reference to Figure 19 in the work.

 7) I don't understand the caption in Figure 19. - Time (ms).

 8) Many editorial errors should be corrected.

9) In the introduction, information on the different arrangements of Arrays should be added

10)  Have the authors considered the influence of temperature change (heating) on the device's operating parameters?

Author Response

Please see the attachment. Thank you for your time.

Reviewer 3 Report

The authors have identified an opportunity to improve the design of magnetoelectric velocity sensors by focusing and increasing the magnitude of the magnetic field supplied by the permanent magnet to the ME structure. The sensitivity of such a sensor is proportional to the field intensity which they show to be increased by almost 30% in their design. The authors do a good job of citing previous works and discussing the downfalls of the current state of the art. Diagrams in figure 6 and 7 are well made and make clear the design being proposed and compares to the conventional design.

I feel that the article is well written and technically sound. In it's current state i feel it could be published without delay.

Author Response

We sincerely appreciate your time and effort to review this manuscript and giving positive comments. Your kind comments give us great encouragement and confidence. Thank you so much! We have double-checked the manuscript so that the typos and grammar errors we found can be corrected. Furthermore, some detailed information was added to make it complete. We have uploaded a revised manuscript and a copy with changes marked in different colors. Looking forward to your comments and sorry for taking up your valuable time.

Round 2

Reviewer 2 Report

The publication has been corrected according to my comments.